# Growth Mode and Physiological State of Cells Prior to Biofilm Formation Affect Immune Evasion and Persistence of *Staphylococcus aureus*

**DOI:** 10.3390/microorganisms8010106

**Published:** 2020-01-12

**Authors:** Kirsi Savijoki, Ilkka Miettinen, Tuula A. Nyman, Maarit Kortesoja, Leena Hanski, Pekka Varmanen, Adyary Fallarero

**Affiliations:** 1Pharmaceutical Design and Discovery (PharmDD) Group, Pharmaceutical Biology, Division of Pharmaceutical Biosciences, Faculty of Pharmacy, University of Helsinki, Viikinkaari 5, 00014 Helsinki, Finland; ilkka.miettinen@helsinki.fi (I.M.); maarit.kortesoja@helsinki.fi (M.K.); leena.hanski@helsinki.fi (L.H.); adyary.fallarero@helsinki.fi (A.F.); 2Department of Immunology, Institute of Clinical Medicine, University of Oslo and Rikshospitalet Oslo, 0372 Oslo, Norway; t.a.nyman@medisin.uio.no or; 3Department of Food and Nutrition, University of Helsinki, 00014 Helsinki, Finland; pekka.varmanen@helsinki.fi

**Keywords:** staphylococcus aureus, growth mode, surfaceome, persistence/tolerance, immune evasion

## Abstract

The present study investigated *Staphylococcus aureus* ATCC25923 surfaceomes (cell surface proteins) during prolonged growth by subjecting planktonic and biofilm cultures (initiated from exponential or stationary cells) to label-free quantitative surfaceomics and phenotypic confirmations. The abundance of adhesion, autolytic, hemolytic, and lipolytic proteins decreased over time in both growth modes, while an opposite trend was detected for many tricarboxylic acid (TCA) cycle, reactive oxygen species (ROS) scavenging, Fe-S repair, and peptidolytic moonlighters. In planktonic cells, these changes were accompanied by decreasing and increasing adherence to hydrophobic surface and fibronectin, respectively. Specific RNA/DNA binding (cold-shock protein CspD and ribosomal proteins) and the immune evasion (SpA, ClfA, and IsaB) proteins were notably more abundant on fully mature biofilms initiated with stationary-phase cells (SDBF) compared to biofilms derived from exponential cells (EDBF) or equivalent planktonic cells. The fully matured SDBF cells demonstrated higher viability in THP-1 monocyte/macrophage cells compared to the EDBF cells. Peptidoglycan strengthening, specific urea-cycle, and detoxification enzymes were more abundant on planktonic than biofilm cells, indicating the activation of growth-mode specific pathways during prolonged cultivation. Thus, we show that *S. aureus* shapes its surfaceome in a growth mode-dependent manner to reach high levofloxacin tolerance (>200-times the minimum biofilm inhibitory concentration). This study also demonstrates that the phenotypic state of the cells prior to biofilm formation affects the immune-evasion and persistence-related traits of *S. aureus*.

## 1. Introduction

*Staphylococcus aureus* is an important Gram-positive pathogen that causes major health problems, especially in the form of recalcitrant infections [1]. Typical examples are infective endocarditis, osteomyelitis, skin and soft tissue infections, and medical device-related infections. These conditions are particularly difficult to manage due to the timely expression of specific virulence factors and the biofilm growth mode. Biofilm formation is an important mechanism employed by bacteria, including *S. aureus*, to sequester scarce nutrients and evade host defenses and antimicrobial chemotherapy [2]. This growth mode is characterized by the aggregation of bacteria into microcolonies, which become engulfed within a primarily self-produced matrix consisting of proteins, extracellular DNA (eDNA), carbohydrate polymers, and/or even host-derived biomolecules. Similar to other biofilm-forming bacteria, *S. aureus* also produces many adhesive proteins that facilitate the attachment of the cells onto host tissue and abiotic surfaces [3,4,5]. In addition to providing mechanical protection and enabling the efficient use of nutrients, the physico-chemically diverse biofilm architecture gives rise to phenotypic heterogeneity within the inhabiting cells [6,7], creating optimal conditions for acquisition of a highly tolerant state known as persistence. Persisters are nongrowing, transiently antibiotic-tolerant bacteria surviving exposure to multiple stresses without undergoing genetic change, which besides in biofilm populations are also present in planktonic cultures [8,9]. Bactericidal antibiotics quickly kill most of a bacterial population, leaving a small fraction of cells that survive by entering a persister state; however, the presence of such survivors is not considered when initiating treatment [10].

Recent studies established that conditions interfering with transcription, translation, or ATP synthesis dramatically increase the persister frequency from 0.01% to 10%–100% [11]. In particular, the cellular depletion of ATP has been found to be a key event for triggering persister formation in *S. aureus*, in which neither the stringent response nor the TA (toxin-antitoxin) system were involved [8]. The cell surface-associated proteins, or surfaceome, are expected to play a role when cells switch into a persister state, as they form the first line of molecular interaction with changes of the environment. To date, the majority of the systematic studies investigating the surfaceome dynamics in *S. aureus* have been conducted with exponential or stationary phase planktonic cells, but cells cultured for prolonged time periods in either planktonic or biofilm states have not been systematically explored by proteomics. As tolerance characteristics outside the immune-evasive barrier are similar among stationary-phase bacteria, planktonic persisters, and biofilm cells [6], we investigated the growth mode-dependent surfaceome changes on *S. aureus* ATCC25923, a control strain used for susceptibility testing [12]. To this end, the cells were prepared in a way that would enhance the formation of persisters during both the planktonic and biofilm growth. As the phenotypic state of the bacterial inoculum can also have a remarkable effect on the resulting phenotypic features of the bacterial cell [13,14], we also investigated whether the physiological stage of the cells preceding biofilm growth affects the surfaceomes in aging biofilms. To the best of our knowledge, this study is among the first to describe and compare surfaceomes from *S. aureus* cultures grown for prolonged time periods in planktonic and biofilm forms.

## 2. Materials and Methods

### 2.1. Bacteria, Media, and Culture Conditions

*S. aureus* ATCC 25923 (HAMBI mBRC—Microbial Domain Biological Resource Centre), a quality control strain in the antimicrobial susceptibility test and a clinical isolate able to form biofilms in vitro [12], was used as the model strain. The strain was routinely cultured on tryptic soy agar (TSA) and in tryptic soy broth (TSB) (Lab M, Lancashire, UK) at 37 °C under aerobic conditions.

Planktonic and biofilm cultures with increased persister-related traits were prepared as follows. For creating biofilm cultures, a loopful (1 µL) of bacterial colonies grown overnight on TSA was transferred into 10 mL of TSB medium in biological triplicates and cells were cultured at 37 °C with shaking (at 220 rpm), for either 2 (corresponding to exponential phase) or 72 h (corresponding to stationary phase). Samples measuring 1.5 mL were transferred into Nunc™ Cell-Culture Treated 24-well plates (Thermo Fisher Scientific, Waltham, MA, USA), and biofilms were allowed to develop for 6 and 24 h (37 °C, 220 rpm). The remaining planktonic cultures continued growth to the comparable time points with the biofilm cultures. At the indicated time points cells were withdrawn and treated as described below for surfaceomic and phenotypic analyses.

### 2.2. Levofloxacin-Susceptibility Analysis

Planktonic and biofilm cells prepared as described above were treated with 400 µM levofloxacin and with mitomycin C (MMC) as follows. Planktonic cells and biofilm cells on 24-well plates were washed once with PBS (phosphate buffered saline, pH 7.2) and then exposed to 400 µM levofloxacin in 2% DMSO (*v/v*)(VWR Chemicals, Radnor, PA, USA), using 2% DMSO as the control. The bacteria were treated for 24 h (37 °C with 220 rpm shaking); next, 500 µL aliquots from planktonic and biofilm cultures (obtained by scraping the cells) were pelleted by centrifugation (8000× *g*, 5 min, 12 °C). Cells resuspended in PBS were serially diluted and plated on TSA in 10-µL droplets. Colony forming units (CFUs) were first determined after an overnight incubation and then after 2 days of incubation to monitor the appearance of slow-growing small colony variants (SCVs). The results were evaluated as reductions in log_10_-transformed CFUs. All treatments were carried out twice using three biological replicates for each. A paired *t*-test was used to compare the means of the samples before and after the treatment. 

### 2.3. Surfaceome Analyses

#### 2.3.1. Trypsin Shaving

The cells were suspended in 100 mM Tris-HCl (pH 6.8); from biofilm cultures cell-samples were obtained by scraping and from the planktonic cultures by centrifugation (3 min, 8000× *g*, +4 °C). Washed cells collected by centrifugation (3 min, 8000× *g*, +4 °C) were suspended gently in 90 μL of TEAB (triethylammonium bicarbonate) containing 10% sucrose [15]. Tryptic digestions were initiated with 55 ng/µL of sequencing grade-modified porcine trypsin (Promega, Madison, WI, USA) and digestions were incubated at 37 °C for 20 min. Released peptides and trypsin were recovered by filtration through a 0.2 µm acetate membrane (Costar^®^ Spin-X Centrifuge Tube Filter, Corning Inc., Corning, NY, USA) by centrifugation (7000× *g*, 3 min, +4 °C). Flow-troughs containing the released peptides and trypsin were further incubated for 16 h at 37 °C. Digestions were stopped by adding trifluoroacetate (TFA) to a final concentration of 0.6% (*v/v*). The peptide concentrations were measured with Multiskan Sky using the µDrop plate (Thermo Fisher Scientific; Vantaa, Finland).

#### 2.3.2. LC-MS/MS and Label-Free Quantification 

The tryptic peptides were concentrated and purified using ZipTip C18 (Millipore) and equivalent quantities of peptides from each sample were submitted to an Easy-nLC 1000 nano-LC system (Thermo Scientific,) coupled with a quadrupole Orbitrap mass spectrometer (Q ExactiveTM, ThermoElectron, Bremen, Germany) equipped with a nanoelectrospray ion source (Easy-SprayTM, Thermo Scientific), as previously reported [16]. The liquid chromatography (LC) separation was performed in an Easy-SprayTM column capillary with a 25-cm bed length (C18, 2-μm beads, 100 Å, 75-μm inner diameter, Thermo Scientific), using a flow rate of 300 nl/min. The peptides were eluted with a 2%–30% gradient of solvent (composed of 100% acetonitrile and 0.1% formic acid) in 60 min. The obtained mass spectrometry (MS) raw files were submitted to MaxQuant software (MaxQuant v.1.6.1.0) [17] for protein identification and label-free quantification (LFQ) using an in-house database composed of *S. aureus* ATCC25923 protein sequences (CP009361 and CP009362) [12] in forward and reverse. Carbamidomethyl (C) was set as a fixed and methionine oxidation as a variable modification. A first search peptide tolerance of 20 ppm and a main search error of 4.5 ppm were used. Trypsin without proline restriction enzyme option was used with two allowed miscleavages. The minimal unique + razor peptide number was set to 1, false discovery rate (FDR) to 0.01 (1%) for peptide and protein identification, and label-free quantitation with default settings was used. Known contaminants provided by MaxQuant and those identified in the samples were excluded from further analysis. Missing value imputation for principal component analysis (PCA) was performed for proteins identified in at least 2/3 replicates in at least one sample group using Perseus [18] v.1.5.6.0 by random draws from a low-abundance-adjusted normal distribution. Statistical comparisons of the LFQ levels were also carried out in Perseus using a *t*-test with a permutation-based false discovery rate of 0.05. Further statistical analyses were performed in IBM SPSS Statistics version 25 (IBM; Armonk, New York, US). Surfaceome patterns were illustrated in Perseus (v. 1.6.2.3) by standardizing (z-score) the log_2_-transformed LFQ abundances across the samples and carrying out complete-linkage hierarchical clustering according to Lp norm distance (*p* = 1.5) both row- and columnwise. The mass spectrometry surfaceomics data are deposited to the ProteomeXchange Consortium via the PRIDE [19] partner repository with the dataset identifier PXD016344.

#### 2.3.3. Proteome Bioinformatics

The proteome was manually curated by characterizing hypothetical and tentatively annotated proteins (when possible) with the aid of the BLAST (Basic Local Alignment Search Tool) program from the NCBI (National Center for Biotechnology Information, Washington, DC, USA) [20,21,22], combined with conserved domain identification using Conserved Domain Database/Subfamily Protein Architecture Labeling Engine (CDD/SPARCLE) [23] and SmartBLAST, UniProt [24] searches. General protein functions were annotated using the Gene Onthology (GO) database [25] and illustrated with the WEGO 2.0 tool [26]. Isoelectric points (pIs) and molecular weights (MWs) were predicted using EMBOSS Pepstats [27,28]. The presence of possible protein secretion motifs (classical and nonclassical) in the identified proteins was predicted with SignalP4.1 [29] and SecretomeP 2.0 [30]. Grand average of hydropathy (GRAVY) values were obtained using GRAVY Calculator. Predictions of transmembrane helices were acquired with the TMHMM Server v. 2.0 [31,32]. The published *S. aureus* ATCC25923 genome [12], carrying 2725 protein encoding genes, was updated to link Gene Ontology (GO) terms for 162 proteins/genes without GO annotation.

### 2.4. THP-1 Phagocytosis Assay

A protocol modified form [33] was used to study *S. aureus* intracellular survival in macrophage-like cells. THP-1 (ATCCTIB202) monocyte cultures were maintained throughout the assay at 37 °C in the presence of 5% CO_2_ in a base medium containing RPMI 1640 (Dutch edition) (Gibco, Invitrogen, Thermo Fisher Scientific, Paisley, UK) supplemented with 10% fetal bovine serum (FBS) (BioWhittaker, Lonza, Basel, Switzerland) and 2 nM l-glutamine (BioWhittaker). The cells were seeded into 24-well tissue culture plates (Costar, Corning Inc., Corning, NY, USA) at a density of 4  ×  10^5^ cells/well in a base medium supplemented with 0.05 mM 2-mercapthoethanol (Gibco) and 20 µg/mL gentamicin (Fluka, Buchs, Switzerland) and differentiated with 160 nM phorbol 12-myristate 13-acetate (PMA, Sigma-Aldrich, St. Louis, MO, USA) for 48 h into macrophage-like adherent cells. Biofilms corresponding to the samples EDBF_24 h and SDBF_24 h were grown and harvested as described above in four individually inoculated replicate cultures. The suspended biofilms were vortexed briefly to disperse the possible remaining aggregates and adjusted to OD_600_ = 0.07 with 100 mM Tris-HCl, pH 6.8. The differentiated THP-1 cells were conditioned in the base medium (in the absence of gentamicin and mercaptoethanol) for 30 min and infected at a MOI (multiplicity of infection) of 1 in a final volume of 500 µL. The plates were centrifuged for 10 min at 500× *g* and further incubated for 50 min. Next, the infected medium was replaced with the base medium supplemented with 100 µg/mL gentamicin and 20 µg/mL lysostaphin and incubated for 1 h to kill the extracellular bacteria. The wells were then washed twice with PBS and the cells were detached with 0.25% trypsin, suspended in the base medium in a final volume of 1 mL, and pelleted by centrifugation at 500× *g* (5 min, 4 °C). The pellets were resuspended in 1 mL of sterile water and incubated for 10 min at 37 °C to lyse the THP-1 cells. Serial dilutions of the lysates were prepared in PBS, plated on TSA in duplicate, and incubated overnight at 37 °C under aerobic conditions to enumerate intracellular bacteria. Additional plates for the further time points were replenished with 1 mL of the base medium containing 20 µg/mL gentamicin and incubated until sampling for 3 and 24 h. An unpaired *t*-test was used to compare differences in bacterial viability. 

### 2.5. RNA Isolation and Droplet Digital Reverse Transcription-PCR (ddRT-PCR)

For relative quantification of the *isaA*, *isaB,* and *hglB* expression by ddRT-PCR, the following primers were designed: *isaA*-for: 5′-AACCTGAAGCACCTGATGGG-3′, *isaA*-rev:5′-CTGCAGGTGCTAC TGGTTCA-3′, *isaB*-for: 5′-CCGCCTGTGCTTCTTGATGT-3′, *isaB*-rev: 5′-GTGCAGCAACGACAT TAGCA-3′, *hlgB*-for: 5′-TGGCGGACTTAACGGAAACA-3′, *hlgB*-rev: 5′-TGTGCTTCTACACCCCA ACC-3′) were designed using Primer3Plus [34] and synthesized by Metabion GmbH. For total RNA isolation, the PL8 h, PL96 h, EDBF_24 h, and SDBF_24 h cells were harvested in RNAlater (Qiagen, GmbH, Hilden, Germany) and the RNA was extracted using the RNeasy Mini Kit (Qiagen, GmbH) with DNase (RNase-Free DNase set; Qiagen, GmbH) treatment according to the manufacturer’s instructions. To remove contaminating DNA, the isolated total RNA was subjected to an additional DNase treatment using the RNase-Free DNase (Qiagen, GmbH) kit followed by an additional purification with RNeasy^®^ Mini Kit (Qiagen, GmbH). First-strand synthesis of cDNA was performed using the iScript Advanced cDNA Synthesis Kit for RT-qPCR (Bio-Rad, Hercules, CA, USA). Equal amounts of RNA samples were used in control reactions without reverse transcriptase. Reverse transcription (RT) reactions were incubated for 30 min at 46 °C and diluted 1:200, and 5 µL was used in ddPCR reactions (22 µL) using 1 × QX200 ddPCR EvaGreen Supermix (Bio-Rad, Hercules, CA, USA). For droplet generation the Automated Droplet Generator (Bio-Rad, Hercules, CA, USA) was used and the thermal cycling protocols were as previously described [35]. The fluorescence of each droplet was measured by a QX droplet reader (Bio-Rad, Hercules, CA, USA), and the results were analyzed using QuantaSoft software (version 1.7.4.0917; Bio-Rad), and normalized to the total RNA used in cDNA synthesis. The ddRT-PCR assay was conducted with two biological replica samples. 

### 2.6. Adherence Assays

The materials for adherence assays were prepared as follows. Briefly, the 96-well Polysorp microplate wells (Nunc Immuno plates, Nunc, Denmark) were treated with 250 μL of PBS or PBS containing the fibrinogen at 200 μg/mL. After overnight incubation at 4 °C the wells were washed twice with PBS, blocked with 2% BSA, or incubated with PBS for 30 min at room temperature. Wells were washed twice with PBS and then allowed to dry. The planktonic cells withdrawn at 8- and 96-h (4 mL from each) by centrifugation, as described above, were washed once in ice-cold Tris-HCL (100 mM, pH 6.8) and then divided in two aliquots; the first half was washed with ice-cold TEAB (100 mM, pH 8.5) and the second was washed with Tris-HCL (100 mM, pH 6.8). After centrifugation (each 3 min, 8000× *g*, +4 °C), the cells were suspended in PBS (2.0 mL). From this suspension 250 μL was added to the fibrinogen-, BSA- and PBS-treated wells, and plates were incubated at 37 °C for 2 h (250 rpm). Non-adherent cells were removed, wells were washed twice with PBS, and adherent cells were stained with 200 μL of the crystal violet solution (0.1%, *w/v*) (Sigma-Aldrich) for 30 min at room temperature. Excess staining was washed twice with deionized H_2_O and the stained cells were suspended in 94% ethanol. The adherence was quantified by recording the crystal violet retained by the cells at 595 nm using the Multiskan Sky reader (Thermo Fisher Scientific; Vantaa, Finland). The results were normalized to cell density, which was measured at A_595_ from each replica cell sample. The adhesion assay involved four biological replicates, each with four technical replicates. The biological replicate means were compared by the paired *t*-test.

## 3. Results

### 3.1. Increasing Levofloxacin-Tolerance by Prolonged Cultivation

*S. aureus* ATCC25923 was subjected to prolonged growth in planktonic and biofilm forms (Figure 1A) to stimulate the formation of persister cells and increase antibiotic tolerance of the cell cultures. Metabolic status and susceptibility to 400 µM levofloxacin was confirmed with planktonic (PL) and biofilm (BF) cells initiated with exponential-phase (EDBF) and stationary-phase cells (SDBF) at each time point. As shown in Figure 1B, levofloxacin treatment reduced the colony forming ability of the PL8 h cells (see sample codes in Figure 1B) by more than 100-fold compared to the control cells (*p* < 0.01). No significant decrease in the CFUs (colony forming units) with PL96 h cells or EDBF and SDBF cells at the 24-h time point was detected. Each planktonic and biofilm cell sample was also treated with 400 μM mitomycin C (MMC) with known antipersister function [36], which reduced the CFUs in each sample below the limit of detection. Thus, the created *S. aureus* cultures were metabolically active and susceptible at the 8-h time point, whereas planktonic and biofilm cells displayed a persister phenotype at later time points of growth.

### 3.2. LC-MS/MS Analyses Generated High-Quality Surfaceome Catalogs

Planktonic and biofilm-associated surface proteins at the indicated four time points (Figure 1A,B) were shaved by trypsin and analyzed by LC-MS/MS. Overall, 847 and 834 proteins from planktonic and biofilm cells were identified, respectively, which are listed in Appendix A. The label-free quantification results were further complemented with physico-chemical and functional data, which are shown in Appendix A. The high quality of the proteomic data is demonstrated by the identification of proteins with at least three or more peptides (75%) having an average sequence coverage of approximately 29% and with only 12% of all proteins being designated as single-peptide hits. In addition, the overlap in protein identifications among the biological replica samples was extensive; 80%–94% of the identified proteins are shared by each replicate (Appendix A).

### 3.3. Similarities and Differences between the Planktonic and Biofilm Surfaceomes

The Venn diagram in Figure 2A compares all planktonic and biofilm-associated surface proteins and shows that over 97% (828 proteins) are shared by the two growth modes. The number of proteins specific to planktonic and biofilm surfaceomes was 19 and six proteins, respectively. Figure 2B compares the number of time point-dependent identifications between the planktonic and biofilm surfaceomes and indicates over 600 commonly identified proteins at each time point and growth mode. The number of biofilm-specific proteins was highest at the 8-h time point, whereas in planktonic cells, the number of specifically identified proteins was highest after 96 h. Figure 2C shows that the number of proteins shared by the majority of the time points was 121 in planktonic cells (PL26–PL96 h). In EDBF cells, the number of specifically identified proteins increased with time, whereas majority of the proteins were shared by SDBF cells at 6- and 24-h time points, and only one protein was specific to both time points. Proteins specific to planktonic and biofilm growth modes are listed in Appendix A. A three-dimensional principle component analysis (3D-PCA) of detected surfaceomes indicated that each replicate samples cluster together, and the tested variables affected the protein abundance patterns (Figure 2D). The first component separates surfaceomes by total age of the cell samples with a strong positive correlation (0.896), whereas PC2 clearly divides the SD-biofilms and the corresponding planktonic cultures to the negative and positive (0.638) ends of the axis respectively. The rest of the surfaceomes arrange between the growth modes according to PC3. Thus, the detected surfaceomes show both the growth mode- and growth stage-dependent changes.

### 3.4. Multivariate Analyses of the Planktonic- and Biofilm-Associated Protein Patterns

The surfaceome abundance patterns were next compared by a heatmap with hierarchical clustering. Figure 3A shows that the surfaceomes of EDBF and planktonic cells at corresponding time points clustered together, whereas the surfaceomes for the SDBF and comparable planktonic cells separated into their own clusters. Six clusters representing the most distinctive patterns are shown; the clusters (i) one and two included cytoplasmic proteins (i.e., transmembrane transport, DNA replication/repair, cell wall biogenesis, and/or cell division) showing higher abundances in older planktonic cell surfaces compared to other cell samples, (ii) three and five contained ribosomal proteins (r-proteins) and pathogenicity-related and other classically secreted proteins with higher abundances in EDBF-, PL8 h- and PL26 h-associated samples, and (iii) clusters four and six had stress response and ROS (reactive oxygen species) scavenging enzymes with higher abundances on SDBF or SD- and PL78 h/96 h-associated cells. All identified proteins from each six clusters were also plotted against their corresponding GO terms, which resulted in the enrichment of eight clearly distinguishable groups of proteins (Appendix A); three in the cellular compartment (cell/intracellular part), four for molecular function (hydrolase/transferase/oxidoreductases, binding of drugs, carbohydrate-derivatives, and small molecules) and one for biological processes (cellular/primary/biosynthetic/small molecule, and nitrogen/organic substance metabolic processes).

In total, 777 moonlighters (including 204 with SecP scores > 0.5) from planktonic and 566 (including 199 with SecP scores > 0.5) from biofilms were detected on cell surfaces. The number of the identified proteins that were predicted to be secreted via the classical (sec-dependent) pathway were 70 and 69 for the planktonic and biofilm cells, respectively. A 3D-PCA investigating the effects of the growth mode and growth stage on the secretion modes shows that the classically secreted proteins are slightly overrepresented (*p* = 0.002) on planktonic cell surfaces at 8- and 26-h time points, and that the component loading of these proteins follows a distinct pattern (*p* < 0.001) (Figure 3B). When compared with the respective growth mode and time point-dependent trends, the classical secretion is suggested to be more efficient in younger planktonic cells and in biofilms initiated with exponential cells (PC1), with higher efficiency in biofilm than planktonic cells. Many of the proteins that load highly positive on PC1 belong to the group of predicted moonlighters. A low but significant (*p* < 0.01) difference between the PC2 and the SecP scores, indicating non-classical function for the proteins, was also detected, suggesting higher abundances of moonlighters on biofilms compared to planktonic cell surfaces.

### 3.5. Classical Virulence and Resistance Proteins

We identified several virulence, adhesion, immune evasion, and resistance-associated factors; Table 1 illustrates a heatmap reflecting their abundances (LFQ intensities) and Appendix A shows the fold-changes for the same proteins between the indicated growth modes and time points. The highest LFQ intensities, indicating high protein abundances, ranged between 30 and 35 (log_2_) and were detected for the immunoglobulin binding protein SpA in PL8 h and SDBF_24 h cells and an NLPA-type ABC transporter (lipoprotein) specific for methionine in SDBF_24 h cells. A decreasing trend in abundances of known virulence factors or adhesins (e.g., coagulase, SasF, Emp, immunoglobulin binding proteins Sbi, SdrD, SdrE-truncated, SdrC, clumping factors ClfA/B, autolysins Sle1, and Atl) was observed over time for both the planktonic and biofilm cells. The greatest decrease was found for Sbi and the truncated form of SdrE. Proteins contributing to peptidoglycan synthesis/resistance such as FemA and FemB were increasingly produced with time, while an UDP-N-acetylmuramoyl-l-alanine-d-glutamate ligase (MurD) displayed a temporal increase only on planktonic cultures at the 26-h, 78-h, and PL96-h time points.

Comparison of the EDBF_6 h and corresponding PL surfaceomes at the 8-h time point revealed that the immunoglobulin binding protein—Sbi, accumulation associated—ClfB, immunodominant antigen—IsaA, and the adhesins—SrdD/E had 30- to 60-fold higher abundances of EDBF when compared to PL cells (Appendix A). The fully matured SDBF_24 h culture and the equivalent PL cells at the 79-h time point were proposed to contain the highest number of persister cells. Comparison of the SD-biofilms at the 6-h time point to the planktonic cells at a comparable time point (PL78 h) indicated >30-times higher abundance for the immuno evasion protein—IsaB and 7–18-fold higher abundances for IsaA, HlgA, and SdrD/E adhesins. Greatest differences among the EDBF/SDBF_24 h- and PL 26 h/96 h-surfaceomes were detected between the SDBF_24 h and PL96 h cells; HlgA, ClfA/B, IsaB, YPEB-like peptidase, ABC transporter CntA displayed 6–13-fold higher abundances on SDBF cells than on PL cells. Thus, these abundance changes suggest that *S. aureus* decreases cell wall modification enzymes over time during both growth modes, whereas in biofilm cells specific virulence, immune evasion, persistence, and/or adhesion-related proteins are produced more with increasing age.

### 3.6. Nonclassical Proteins Relevant to Virulence and Resistance

Appendix A lists the identified known and predicted moonlighters together with their relative abundance differences between the indicated conditions. Among these, the largest protein group was the 30 S and 50 S r-proteins. Other known moonlighters, including the chaperone DnaK, enolase, elongation factor (EfTU), superoxide dismutase (SodA), trigger factor (TF), an alkyl hydroxyperoxidase (AhpC/F), pyruvate kinase (PYK), phosphoglycerate kinase (PGI), phospho-fructokinase (PFK), phosphoglycerate kinase (PGK), malate-quinone oxidase (MQO), fructose-bisphosphate aldolase (FBA), DNA-directed RNA polymerase beta subunits, glyceraldehyde-3-phosphate dehydrogenase (GaPDH), inosine 5’-monophosphate (IMP) dehydrogenase, alkaline shock protein 23, and alkaline shock response membrane protein (AmaP) were also present in high abundances on both the planktonic and biofilm cells. The alkaline shock proteins were differentially produced in EDBF and SDBF cells; both proteins were more efficiently produced (>3-fold) in EDBF than in SDBF cells at the 24-h time point. 

#### 3.6.1. Differences between the PL8 h and PL96 h Surfaceomes

Proteins specific to PL96 h cells or more abundant (4- to 80-fold) on PL96 h than PL8 h cells (Appendix A) included the fatty acid degradation enzymes (acetyl-CoA C-acyltransferase/FadA, long-chain fatty acid-CoA ligase/FadE, glutaryl-CoA dehydrogenase/FadD propionate CoA-transferase/FadX), TCA cycle (2-oxogluatarate dehydrogenase/2-OGDH, aconitase/AcnA, malate dehydrogenase/quinone/MQD), Fe-S repair (SufB), urea cycle (argininosuccinate synthase/AsS, argininosuccinate lyase/AsL, arginine deiminase/ArcD), oligopeptide uptake (OppD), peptidases (PepP, PepM, PepS and PepF, ClpP), hydrolases, stress chaperones (DnaK, DnaJ, GrpE, GroES, DnaJ, ClpB), ROS scavenging enzymes (superoxide dismutases SodM, SDR family NAD(P)-dependent oxidoreductases, catalase, alkyl hydroperoxide reductase subunit F/AhpF), and a DNA translocase (FtsK). Glycolytic moonlighters such as the phosphoenolpyruvate carboxykinase (PCK) and glucose-6-phosphate dehydrogenase (GPD) were more abundant on PL96 h than PL8 h cells. From these identifications the Fad-proteins are encoded by the *fadABDEX*, and the operon and stress response proteins through the *dnaJ-dnaK-grpE* operon. Appendix A shows that in most cases the identification intensities for each protein follow the same trend under the same condition. 

Proteins specific or more abundant (4- to 36-fold) on PL8 h cells compared to PL96 h cells included a hypothetical protein (YfjT), a sperm-coating glycoprotein (SCP)-like extracellular protein, and several phage proteins (a phage major capsid protein, phage-related phi PVLORF 30 analogue, and phage tail protein). Among these, the 6-kDa hydrophilic/acidic YfjT protein was predicted to be the most abundant. Other more abundant proteins on PL8 h cells included the cold-shock proteins CspA and CspD with DNA-binding ability, a penicillin-binding protein 2 p, and an oligopeptide transport protein A (OppA). Thus, ageing planktonic cells have more peptidolytic/hydrolytic, ROS scavenging, and secondary carbon source exploiting activities, while phage- and cold-shock-related functions are more dominating on younger planktonic cells.

#### 3.6.2. Differences between the EDBF_24 h- and SDBF_24 h Surfaceomes

Proteins specific or more abundant (>4-fold) on EDBF_24 h than on SDBF_24-h cell surfaces (Appendix A), included r-proteins (RpsK and RpsU), phage proteins (an infection protein, a major capsid protein, and phage tail protein), esterases (carboxylesterase/lipase family protein, glycerol-phosphodiester phosphodiesterase/GlpQ), an urea cycle enzyme (ArcA), and cold-shock protein (CpsC). 

Specific or more abundant proteins (4- to 25-fold) on SDBF_24 h cells included *i.a*., RNA and DNA binding proteins (r-protein S1 and CspD), stress chaperones/proteases (Zn-metallopeptidase, a membrane-anchored ClpP, ClpC, ClpB, and DnaK), redox/detoxification/ROS scavenging enzymes (Glx, quinol oxidase subunit 2, two thioredoxins, a thiol peroxidase, SodM), a lipoteichoic acid synthase S (LtaS), the *fad* operon gene products (FadA/B/D/E), OppA, and the signal transduction protein TRAP. From these the Fad-proteins, CspD and OppA displayed the greatest abundance increase (>10-fold). Thus, the EDBF cells have more phage-related functions and responses to cope with acidity and nutrient depletion, whereas SDBF cells are more efficient in fatty acid degradation/biosynthesis, uptake of peptides, ROS detoxification, and controlling protein aggregation.

#### 3.6.3. Growth Mode-Dependent Abundance Changes within Commonly Identified Moonlighters 

The most abundant identifications on EDBF_24 h cells included *i.a*., a 7-kDa hypothetical protein, GroES, a vicinal oxygen chelate (VOC) domain protein, a cold-shock protein CspC, polyisoprenoid-binding protein, and a phage infection protein (Yhge/Pip-domain). On SDBF_24 h cells, an YfjT (6 kDa), a 6-kDa hypothetical, the polyisoprenoid-binding protein, a SCP-like extracellular protein, and r-protein S1 were the most abundant. Twenty proteins displayed >4-fold increase in abundances on SDBF_24 h cells compared to their counterparts on PL96 h cells. These included CspD with 102-fold, CspA with 15-fold, OppA with 70-fold, and penicillin-binding protein 2 p with 14-fold higher abundances. D-alanyl-lipoteichoic acid biosynthesis protein D (DltD) was >3-fold more abundant on SDBF_24 h- than on PL96 h cells.

Proteins specific or more abundant on planktonic cells included the phenol-soluble modulin export protein C (PmtC), an UDP-N-acetyl muramoyl-l-alanine-d-glutamate ligase—MurD, Fic, and fructosamine kinase family proteins. Thus, these findings indicated clear growth-mode-dependent surfaceome changes over time in *S. aureus.* The detected changes suggest that persister-enriched planktonic and biofilm cultures differ in terms of phage-related functions, and specific virulence-, stress-, and resistance-associated factors.

### 3.7. In Vitro Viability of Fully Matured Biofilm Cells (EDBF_24 h, SDBF_24 h)

The identification data implies that the EDBF and SDBF_24 h cells differ in terms of their immune evasive features (e.g., SpA, Sbi, ClfA, and IsaB). To test this possibility, the fully mature biofilm cells were subjected to intracellular killing using phagocytic mammalian THP-1 macrophage-like cells. The viability of the intra-cellular bacteria monitored at one, three, and 24 h post-infection (hpi) indicated no significant difference in intracellular bacterial load between the ED- and SD-biofilms at 1 hpi (9.88 × 104 ± 2.28 × 104, *t*-test *p* = 0.19) or at 3 hpi (*p* = 0.17 and *p* = 0.59 for EDBF and SDBF cells, respectively). At 24 hpi, a major reduction in the survival of both the EDBF and SDBF cells was detected (Figure 4A).

### 3.8. RT-ddPCR Analyses of the isaB, isaA, and hlgB Genes

An additional RT-ddPCR analysis for genes coding IsaB (with 59-fold higher abundance on SDBF_24 h than EDBF_24 h), IsaA (with 32-fold higher abundance on PL8 h than PL96 h cells), and HlgB (WP_000595617.1) with 61-fold higher abundance on PL8 h than PL96 h cells was conducted to compare transcriptional and post-translational level abundances. Figure 4B indicates the relative abundance of each transcript in PL8 h, PL96 h, EDBF_24 h, and SDBF_24 h cells. Among these, the relative abundance of *isaB* transcripts were the highest in planktonic cells at both time points, and the lowest in biofilm cells. The *isaA* transcripts displayed a time-dependent decrease in planktonic cells, whereas in biofilm cells the relative *isaA* abundance was higher in SDBF than in EDBF_24 h cells. The highest level for the *hlgB* transcripts was observed in PL8 h cells and the lowest was in PL96 h and the biofilm cells. Figure 4C compares the transcriptional and protein abundance levels from the PL8 h and PL96 h cells. Between the metabolically active PL8 h and levofloxacin-tolerant PL96 h cells, IsaA and HlgB demonstrate parallel reduction in both the protein abundance and corresponding transcript abundance. For IsaB the protein abundance increased towards the 96 h time point, whereas the corresponding transcript level was either constant or only slightly increased towards the end of the growth in planktonic cells. In biofilm cells, the relative amounts of *isaB* and the corresponding protein abundance changes in biofilm cells are markedly different; IsaB abundance increased sharply from EDBF to SDBF_24 h biofilms, a change that was not detected at the transcript level (Figure 4D). Thus, the transcription and protein expression profiles correlated in planktonic but not in biofilm cultures.

### 3.9. Adherent Features of Planktonic Cells at 8- and 96 h Time Points

Bacterial adherence to medical surfaces is the first step of a successful infection, in which the cell surface hydrophobicity plays a critical role [37]. Here, we wished to explore this feature by comparing the combined LFQ intensities of all identified planktonic and biofilm proteins (representing total abundances of all proteins) with GRAVY values exceeding 0.00 (increasing hydrophobicity) and below 0.00 (increasing hydrophilicity) at different time points during growth. Figure 5A (upper panel) shows that the total protein-mediated hydrophobicity decreases over time in planktonic and EDBF proteins, whereas an opposite trend is observed for the SDBF proteins. The combined LFQ intensity of the EDBF_24 h-associated identified proteins are comparable with the combined LFQ intensity of all identified proteins from SDBF_6 h cells. Similar trends were also detected for hydrophilic proteins (Figure 5A, lower panel). However, the total abundance of the non-classical surface proteins increased with time on both the planktonic and biofilm cells, as demonstrated by the PCA above. As the cells preceding biofilm formation are in planktonic form, we next tested the ability of the PL8 h and PL96 h cells, showing the greatest difference in protein-mediated hydrophobicity to bind hydrophobic material and human fibronectin. Figure 5B (upper panel) shows that the relative adherence of non-washed PL8 h cells to hydrophobic surface is approximately 25% more efficient (*p* = 0.00021) compared to PL96 h cells. Alkaline washing, a method known to release surface-associated moonlighters [5,36,37,38] from cells, increased the adherence of PL8 h cells to the hydrophobic surface, whereas no significant effect between the non-washed and washed PL96 h cells on the same material was observed. Figure 5B (lower panel), in contrast, reveals that the ability of the same cells to bind human fibronectin was slightly higher with PL96 h cells than PL8 h cells, and washing the cells with a pH 8.5 buffer resulted in a slight decrease (*p* < 0.05) in cell adherence to fibronectin.

Figure 5C indicates that both the PL96 cells and the PL8 h cells have proportionally more acidic and hydrophilic surface proteins compared to the alkaline (upper panel) or hydrophobic (lower panel) surfaceomes of the same cells. The most abundant proteins included a hydrophilic SpA and an alkaline dipeptide lipoprotein transporter. A membrane-associated ClpP, a short-chain dehydrogenase/reductase (SDR) and a phosphotransferase system (PTS) transporter IIBC subunit were among the most abundant hydrophobic proteins on the PL96 h cells, whereas an alkaline protein (WP_000267034.1) was detected over 30-fold more abundant on the 8-h cells than 96-h cells. Thus, these findings imply that classical and non-classical surface proteins could have a strong effect on the adherent features of planktonic *S. aureus* cells.

## 4. Discussion

### 4.1. Levofloxacin-Tolerance is Effectively Formed in Aging Staphylococcal Cultures

Despite considerable research efforts, antibiotic resistance is increasing and is an important cause of anti-infective therapy failure. In this study, the surfaceome dynamics in planktonic and biofilm cells cultured for prolonged time periods (96 h in total) were investigated using *S. aureus* ATCC25923 as the model. As the most efficient persister formation in planktonic cultures can be triggered using stationary-phase cultures as inocula [38,39], we wished to compare if the antibiotic tolerance/persistence of the cells is affected when 2 h and 72 h old planktonic cultures were directly transferred, without diluting into fresh medium, into conditions enabling biofilm formation. The increased tolerance and the presence of possible persister cells was validated by exposing the resulting planktonic and biofilm cells to 400 μM levofloxacin. Many conventional antibiotics, such as beta-lactams, fail to kill any non-growing bacteria (persisters) regardless of other persistence-related traits. Fluoroquinolones, on the other hand, are more broadly active against non-growing bacteria due to their ability to interfere with DNA replication. Levofloxacin that is a fluoroquinolone antibiotic was therefore chosen to monitor changes in antimicrobial tolerance, which arises not only from ceased growth and cell division, but also from a more complex and extensive phenotypic alteration [40]. For levofloxacin, a minimum biofilm inhibitory concentration (MBIC) of 1.38 µM had been previously established [41]. Thus, the oldest planktonic and fully matured biofilm cells tolerated levofloxacin at concentrations clearly exceeding the MBIC (>200 times). This is in line with the notion that bacteria in dense populations, such as biofilms, or in the stationary phase display increased tolerance to antibiotics compared to exponentially growing planktonic cells [42,43,44,45] and that persisters are effectively formed by *S. aureus* during the stationary phase of growth phase [8].

### 4.2. Cell Wall-Modifying Mechanisms Depend on the Growth-Mode 

Among the planktonic and biofilm cultures, only the youngest planktonic cells (PL8 h) displayed a reduced ability to form colonies after an overnight treatment with levofloxacin. Surfaceome differences between the PL8 h and levofloxacin-tolerant planktonic cells at later time points and each biofilm culture could explain this; several proteins attacking peptidoglycan, including the immunodominant antigens IsaA, SsaA, Atl, and Sle1, which are known enzymes able to hydrolase peptidoglycan, displayed the highest activity on PL8 h cells. Increased tolerance of the PL96 h cells and the biofilm cells at each time point (6- and 24-h) could be due to slightly increased temporal abundance of specific peptidoglycan-strengthening enzymes, such as the peptidyltransferases FemX, FemA, and FemB. These enzymes are responsible for assembling the pentaglycine cross-bridges of the peptidoglycan in *S. aureus* [46]. Peptidoglycan also contains d-alanine and d-glutamate, which contribute to the architecture of the peptidoglycan and provide resistance to most known proteases [47]. Our study identified MurD, a UDP-N-acetylmuramyl-l-alanine-d-glutamate ligase as more abundant on planktonic cultures, with the highest level being reached on PL96 h cells. This enzyme catalyzes the addition of d-glutamate to the nucleotide precursor UDP-N-acetyl muramoyl-l-alanine and increased d-alanylation of the lipoteichoic acids (LTAs) has been shown to increase daptomycin (DAP) tolerance in *S. aureus* [48]. LTAs also have important roles in maintaining cell wall integrity, involving the regulation of cell autolysis and coordinating cell division and virulence [49,50]. The overexpression of lipoteichoic acid synthase (LtaS) and the associated D-alanylation enzymes can contribute to DAP tolerance in methicillin-resistance (MRSA) in *S. aureus* [51]. In this instance, LtaS was notably more abundant on planktonic than on biofilm cells, while the accompanying d-alanylation enzymes (d-alanine-poly/phosphoribitol ligase, d-alanine amino-transferase, and d-alanine-d-alanine ligase) displayed slightly increasing abundance trends during both growth modes.

Penicillin-binding proteins, having both trans-glycosylase and trans-peptidase activities, are involved in catalyzing the last stages in peptidoglycan biosynthesis, which is necessary for the elongation of glycan chains and the formation of peptide bonds, respectively [52]. Here, our findings indicated that the synthesis of the penicillin-binding protein 2 p, while decreasing over time in both the planktonic and biofilm cells, is approximately 14-times less produced during planktonic than biofilm growth after 96 h. Thus, our findings indicated that similar cell wall-strengthening mechanisms are used by both growth modes to increase viability under prolonged cultivation, but specific abundance differences could be necessary for planktonic and biofilm cells to survive under these two conditions. 

### 4.3. Growth Mode and Growth Stage Affects Classical and Non-classical Protein Export

Many classical virulence factors and surface adhesins decreased over time during both growth modes, which agrees with earlier reports showing that cells at exponential growth are more virulent than at the stationary phase [53]. Such metabolic change seems reasonable, as cells need to switch on the energy-saving growth mode by reducing the synthesis of proteins not needed during prolonged growth. On the other hand, the total abundances of all classical surface proteins increased over time in both growth modes, whereas total abundances of nonclassical surface proteins, including known and predicted new moonlighters, displayed growth mode-dependent change. The abundance of moonlighters decreased over time during planktonic growth, whereas an opposite trend was observed for biofilms. It is currently believed that the release of moonlighters depends on regulatory lysis caused by increased autolysin, phenol-soluble modulin-mediated activities, and/or by decreased peptidoglycan crosslinking [54,55,56,57,58,59]. Since autolysins (Atl and Sle1) and phenol-soluble modulins (PmtC) were identified on PL8 h, and/or EDBF_6 h and SDBF_6 h cells, we suggest that such activities contributed to the release of moonlighters in the present study. Here, majority of the cell-surface proteins were predicted as moonlighters (86% of all planktonic proteins, 76% of all biofilm proteins) with likely roles in glycolysis, the TCA cycle, stress, translation, or ROS detoxification. It has been proposed that a specific selection must take place to select which cytoplasmic proteins are exported out of the cells as no correlation between the quantity and abundance levels of moonlighters have been yet been indicated [56,57,58]. Our findings support this by showing that the total abundances of the identified moonlighters show different trends during planktonic and biofilm growth, despite the similar expression patterns for autolysins and phenol-soluble modulins in both cell types. 

### 4.4. Planktonic and Biofilm Cells Use Similar and Distinct Pathways for Increasing Resistance

#### 4.4.1. TCA- and Urea-cycle Mediated Effects

The tricarboxylic acid (TCA) cycle is a central metabolic pathway for generating energy (ATP) and precursors for the biosynthesis of macromolecules, such as oxaloacetate, 2-oxoglutarate, succinate, malate, fumarate, isocitrate for coordinating acetate and amino acid catabolism. Increased TCA cycle activity is advantageous for maintaining viability under nutrient- and oxygen-limited conditions in *S. aureus* biofilms [60], while decreased TCA cycle activity has been linked with increased persister formation during the stationary phase [61]. Decreased TCA cycle activity reduces the available pool of NADH, which by preventing the formation of ROS leads to increased survival [62]. On the other hand, TCA cycle-mediated production of ROS can also trigger the activation of SOS mutagenesis increasing viability via beneficial genetic mutations [63]. In this instance, the abundances of many TCA cycle enzymes (AcnA, MDH, MQO, and 2-OGDH), ROS detoxification enzymes (SodA/M, AhpC/F, KatA, and TrxA), and Fe-S repair proteins (SufB/D/C) were increased on cell surfaces during both growth modes. This finding implies that the production of ROS could be induced and that 2-OGDH (producing NADH to provide electrons for the respiratory chain) and aconitase, AcnA (catalyzing the isomeration of citrate to iso-citrate), could have been affected under these conditions; both of these enzymes are highly vulnerable to ROS [62]. However, the increased abundances of these moonlighters on the cell surfaces may not necessarily correlate with TCA cycle activity inside the cells. A recent study showed that decreased TCA cycle activity results in decreased protein levels in the biofilm matrix [64]. In this instance, the predicted total protein abundances decreased only on planktonic cells and biofilm cells initiated with exponentially growing cells over time, while an increasing abundance trend was detected for biofilms initiated with stationary phase cells. Thus, it remains to be shown if the detected TCA cycle changes on the cell surfaces of the planktonic and biofilm cells are in line with the TCA cycle changes occurring in the cell cytoplasm of these cells. 

#### 4.4.2. Responses to Cope with Acid-induced Cell Death

A TCA cycle-associated enzyme, pyruvate oxidase (CidC) was also decreasingly produced over time in both growth modes. *S. aureus* address the lethal effects of CidC by diverting the carbon flux towards neutral rather than acidic by-products (acetate) by enhancing the synthesis of acetolactate synthase (AlsS) and acetate decarboxylase (AlsD) to convert pyruvate to acetoin [65,66]. In this instance, the decreased abundances of CidC and the lack of both Als enzymes imply that the CidC-dependent pathways leading to cell death are not activated or are less active. The urea cycle and arginine metabolism can also coordinate the survival of *S. aureus* in the presence of acetate by neutralizing the acid with ammonia released during arginine catabolism [67]. In this study, we identified related enzymes, such as the arginine deiminase (ArcA), argininosuccinate lyase (ArgH), and arginosuccinate synthase (AsS), as more efficiently produced during planktonic than biofilm growth. Among the two biofilm cultures (EDBF and SDBF), the urea cycle enzymes were detected as more abundant on biofilms initiated with exponential-phase cells than stationary-phase cells. Thus, an efficient arginine metabolism associated with planktonic growth and biofilm growth during early stages could be necessary to cope with acid-induced cell death.

#### 4.4.3. Stress Moonlighters and Tolerance

The cold-shock protein, CspD, with DNA replication inhibitor activity was remarkably more abundant on biofilms derived from stationary phase cells (SDBF_24 h) compared to planktonic (>100-fold) or fully mature biofilms (>15-fold) derived from exponential cells (EDBF_24 h). The function of this protein is closely linked with stringent response (ppGpp and cyclic AMP—cAMP) mediated activities in *E. coli* [68,69]. CspD is induced in the stationary phase or upon carbon starvation and increases the production of persister cells [70]. Recently, this DNA-binding protein was also proposed to play a role in the postantibiotic recovery of *E. coli* [71]. An RNA chaperone, CspA, was also efficiently produced during both growth modes, with somewhat higher abundances on biofilm surfaces. Inactivation of the corresponding gene in *S. aureus* has been shown to increase aggregation and diminish resistance to oxidative stress [72]. A third cold-shock-associated protein, CspC, with implications in virulence and cellular aggregation [73], was also more abundant on fully mature ED and SD biofilms. Interestingly, two r-proteins, S1 and S4, were more abundant or specific on SD-biofilms at 24-h time point compared to ED-biofilm cells or planktonic cells at comparable time points. An RNA-binding protein was recently reported as a classically secreted protein in *Listeria monocytogenes* affecting virulence, which binds a subset of RNAs, leading to their accumulation in the extracellular milieu and the non-self-RNA innate immunity sensor, potentiating interferon-β production [74]. We also identified other moonlighters, such as DnaK-, ClpB-, ClpC-, and ClpP, with known protein-folding/refolding activity; inactivation of the corresponding genes has been shown to decrease the persister fraction in *S. aureus* [75,76]. The association of ClpP with ClpC to form an active protease targets the TA systems in *S. aureus*, which resulted in enhanced persistence of the cells [76,77]. Thus, the presence of Csp-family proteins and stress-related chaperones and/or Clp protease(s) could be the major persistent driving force for *S. aureus* during biofilm growth.

#### 4.4.4. Peptidolytic Moonlighters and Tolerance

Our results also indicated that temporal abundance increases for many proteolytic components, involving peptidolytic enzymes (PepD, PepV PepF, PepP, PepM and PepS) and oligogeptide transporters (OppD) on planktonic cells, with possible roles for providing amino acids for growth. The proteolytic system could have also provided branched chain amino acids induction of persistence. In many bacteria, the (p)ppGpp levels are regulated by an alarmone synthetase/hydrolase Rel in response to amino acid starvation, in which the branched-chain amino acids, such as valine and isoleucine, play an important role [78]. Previously, we demonstrated that in *Lactobacillus rhamnosus*, the surface-associated aminopeptidases (PepN/PepC) degrade oligopeptides while bound to the biofilm matrix, indicating that the cytoplasmic enzymes can also confer a similar function at the cell surface [79]. While the peptidolytic enzymes were less abundant on biofilm cells, an oligopeptide transporter protein, OppA, was notably more strongly produced on fully mature biofilm cells (SDBF_24 h) compared to planktonic cells or biofilm cells derived from exponential-phase cells (EDBF_24 h). In *S. aureus*, OppA has been reported to import nickel and cobalt in zinc-depleted conditions and contribute to virulence [80]. This moonlighter can also bind several host factors, including plasminogen, fibrinogen, laminin, keratin, hyaluronan, heparin, and type I collagen [81]. Thus, it could be that the components of the proteolytic system, involving either the oligopeptide uptake systems, proteases, or peptidases, show different functions depending on the growth mode of the bacterium. 

### 4.5. Surfaceome Changes Contributing to Virulence and/or Immune Evasion 

The identified classical surface proteins with implications in virulence and/or immune evasion included IsaB, SpA, Sbi, ClfA, FnbA, SasF, hemolysins, leukocidins, AtlA autolysin, a YPEB-lipoprotein peptidase, PrsA foldase, and ABC transporters for methionine and manganese. From these proteins, only IsaB, SpA, ClfA, and AtlA displayed a remarkable increase in SDBF cultures at 24-h time point, implying that the SDBF cells could be more immune-evasive than other biofilm or planktonic cells, which was also demonstrated by comparing the cell viability of the SDBF- and EDBF-24 h cells in THP-1. It has been shown that *S. aureus* strains with high SpA content are more resistant to phagocytosis by human neutrophils in vitro than strains with less Spa [82]. Similarly, the increased abundance of the immune-evasion and immunodominant antigen IsaB could have also conferred increased viability to the SDBF_24 h cells, as this protein has been shown to prevent autophagic flux in macrophages [83]. The abundance patterns of SpA and IsaB differ between the EDBF and SDBF cultures, indicating that the phenotypic state of the cells preceding biofilm formation is important for the development of immune-evasion features in the *S. aureus* strain. It is also notable that the *isaB* transcript levels were not in line with the changes in protein abundance. A likely explanation could be that IsaB levels are regulated at the posttranslational level and by the protein stability that could be increased by eDNA binding to the protein [83,84], making the protein more resistant towards proteolytic turnover during prolonged growth. Finally, the accumulation of the associated protein ClfA could have also played a role in immune evasion, as *S. aureus* deficient for this protein is reported to be significantly attenuated in infection in vivo models [85,86]. Taken together, these findings imply that both the growth mode and the phenotypic state of the cells prior to biofilm growth have a clear effect on the immune-evasion traits in *S. aureus* during prolonged growth. Whether these phenotypic changes affect the immune-evasion traits of other *S. aureus* strains or other Staphylococcus species remains to be shown. 

### 4.6. Surfaceome Changes Affecting the Initial Adherence of Planktonic Cells

In this study, the biofilms were formed on a hydrophilic surface, representing the accepted static model for biofilm formation. However, adherence to hydrophobic materials is more clinically relevant as medical implants are hydrophobic in nature (silicon, stainless steel, Teflon) [37]. The hydrophobicity of the bacterial cell plays a key role in mediating the adherence and, thereafter, the adherent growth on hydrophobic surfaces as more hydrophobic cells are more efficient in biofilm formation. Our results indicate that biofilms prepared from stationary phase cells were more immune-evasive than biofilms prepared from exponentially growing cells, but planktonic cells at the 8-h time point, representing cells at the mid-log or late exponential phase of growth, were more adherent to hydrophobic surfaces than late stationary-phase cells. In this instance, the combined LFQ intensities (i.e., comparable to total protein abundances) of all proteins, including both the hydrophobic and hydrophilic proteins, decreased over time on planktonic cells. On the other hand, an opposite trend was detected only with biofilms initiated with stationary-phase cells, whereas biofilms derived from exponentially growing cells displayed a similar decreasing abundance trend with planktonic cells. As the majority of these changes were associated with the presence of moonlighters, it could be that moonlighters were more effectively released towards the end of the growth in EDBF cultures than in SDBF cultures. This process may involve more efficient production of alkaline shock proteins in EFBF than SDBF cultures or more efficient autolysin (AtlA) activity on SDBF than EDBF cells. Alkaline conditions are known to stimulate the release of moonlighters from the cell surfaces [87], and increased abundances of alkaline shock proteins on EDBF cells imply that the pH of the culture medium increases during growth, which subsequently release moonlighters.

Unlike the moonlighters, the total abundances of all classical surface proteins increased with time in both growth modes. The PL8 h cells were more adherent on hydrophobic material than PL96 h cells, which is in line with the decreased hydrophobicity predicted for these cells. On the other hand, PL96 h cells were slightly more adherent to human fibronectin than PL8 h cells, which could be explained by the increased abundances of the classical fibronectin binding proteins, such as AtlA and ClfA, which were more abundant on the PL96 h cells than PL8 h cells. Many moonlighters, such as Ef-TU, enolase, GaPDH, pyruvate dehydrogenase (PDH), and OppA identified in this study can also bind fibronectin [88]. Therefore, we also compared the adherence of the planktonic cells after an alkaline wash as moonlighters show pH-dependent interaction with the cell surface of bacterial cells and treating cells with an alkaline buffer has been shown to release proteins from the cell surfaces [88]. Our results indicated that an alkaline wash affected the hydrophobic interactions only for the PL8 h cells, whereas binding of both PL8 h cells and PL96 h cells to fibronectin was reduced by alkaline washing. Thus, it could be that reduced fibronectin binding was due to the release of specific moonlighters or the release of eDNA bound to certain surface-anchored proteins that can also associate with the biofilm-matrix in a pH-dependent manner [87,88,89]. A recent study of *S. aureus* indicated a strong positive charge of alkaline virulence factors and r-proteins trapped in an acidic biofilm matrix environment [5]. In that study, these proteins were proposed to mediate the electrostatic interactions with negatively charged surface components and eDNA for enhancing aggregation and biofilm integrity. In our study, the r-proteins were detected in each condition with somewhat higher abundances on EDBF-, PL8 h- and PL26 h-associated cells. Nevertheless, the increased hydrophobicity predicted for the SDBF proteins over time implies that a similar mechanism could be exploited to maintain biofilm stability during prolonged growth. In the case of planktonic cells, the detected temporal decrease in hydrophobic interactions is clinically relevant, as initial adherence prior to biofilm formation involves single cells in a planktonic form. 

## 5. Conclusions

The present study revealed both the growth mode- and growth stage-dependent surfaceome changes in levofloxacin-tolerant *S. aureus* ATCC25923 cells grown for prolonged time periods. The resistant planktonic cells were predicted to enhance cell wall strengthening activities and proteolysis/ peptidolysis to release amino acids for growth or development of persistence during prolonged growth. Maturing biofilms were suggested to exploit distinct mechanisms, involving the immune evasion, cold-shock, DNA, and/or RNA binding activities, for increasing resistance and persistence. The TCA-cycle, the CidC-dependent cell death, and ROS scavenging activities exhibited similar abundance trends during both growth modes, implying that these pathways could promote the viability by diverting the carbon flux towards neutral rather than acetate end products and/or by generating beneficial mutations. It could be that the surfaceome changes involving specific moonlighters do not directly correlate with their protein synthesis changes in the cytoplasm. For example, the key TCA enzymes, AcnA and 2-OGDH, could have been increasingly exported from the cells to maintain proper NADH/NAD balance for preventing ROS-mediated cell death. While exporting certain cytoplasmic proteins out of the cells could be an energetically favorable way for *S. aureus* to improve viability under hostile conditions, further studies are needed to confirm this possibility. Nevertheless, cells from biofilm cultures initiated with stationary-phase cells exhibited higher in vitro viability than cells from biofilms initiated with exponential cells. We also demonstrated that exponentially growing planktonic cells are more adherent to hydrophobic surfaces than stationary phase cells that were more adherent to human fibronectin. This indicates that the physiological state of the cells prior to biofilm formation plays a role in clinical settings. In addition, our study demonstrated that the phenotypic state of the planktonic cells prior to adherent growth can have a great impact on the viability of the cells, thereby contributing to persistence. These findings together with the identification results suggest that both classical surface proteins and moonlighters are needed to increase the immune-evasive and persistent traits in *S. aureus*. Taken together, this study is among the first reports describing growth mode-dependent surfaceome changes in aging *S. aureus* cells, demonstrating that distinct metabolic activities are exploited in biofilm and planktonic cells to achieve resistance. 

## Figures and Tables

**Figure 1 microorganisms-08-00106-f001:**
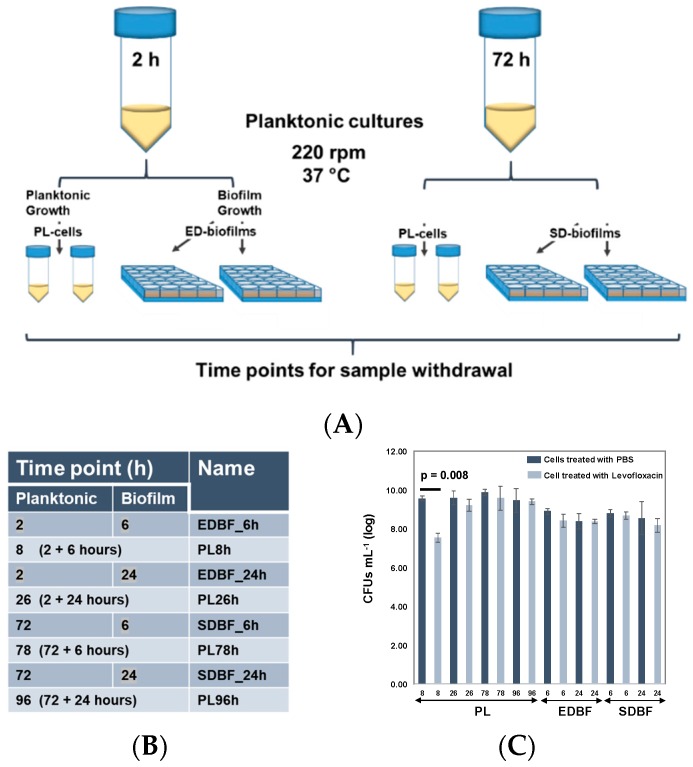
(**A**) Workflow used for preparing the planktonic (PL) and biofilm (BF) cell cultures of *S. aureus* ATCC25923. EDBF refers to biofilm culture initiated with planktonic cells at exponential growth phase (initiation culture propagated for 2 h). SDBF refers to biofilm culture initiated with planktonic cells at stationary growth phase (initiation culture propagated for 72 h). (**B**) Time points of sample withdrawal for surfaceomics and phenotypic analyses. All samples were prepared in biological triplicates. (**C**) Susceptibility of the PL, EDBF, and SDBF cells at the indicated time points was tested with 400 μM levofloxacin. The error bars denote the standard deviation of the log_10_-transformed bacterial concentrations from three individual experiments. The difference between the means of treated and nontreated PL8 h cells is statistically significant (a paired *t*-test). The error bars indicate standard deviation (*n* =16).

**Figure 2 microorganisms-08-00106-f002:**
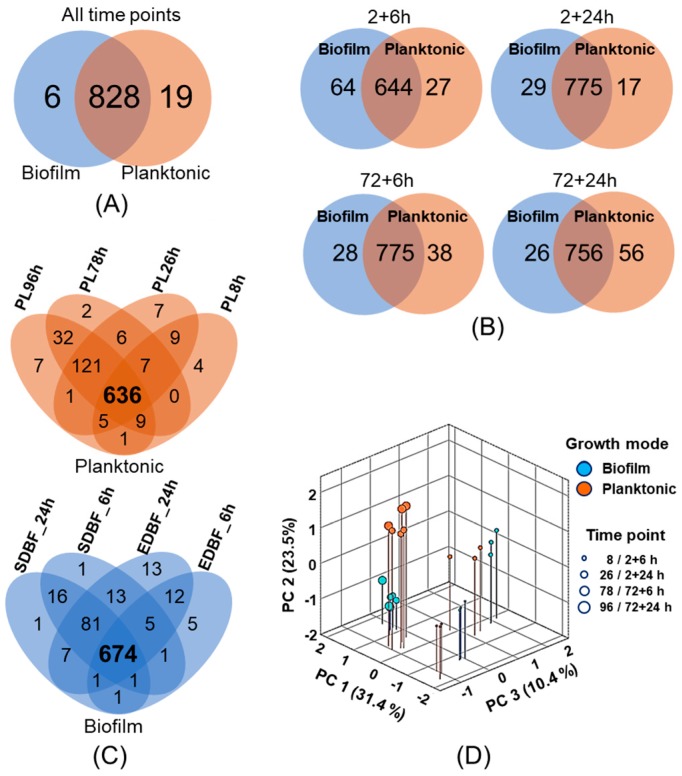
(**A**) Venn diagram comparing the number of all identified proteins from biofilm and planktonic cell surfaces. (**B**) Comparison of the number of specific and commonly identified proteins between the planktonic and biofilm cells at different time points of growth. (**C**) The specific and commonly identified proteins within planktonic and biofilm cell samples at different time points of growth. (**D**) 3D-PCA score plot (varimax rotation and Kaiser normalization) of the individual samples with 3 replicates each. An initial oblimin rotation with Kaiser normalization was pre-applied to verify the absence of intercomponent correlation (|r| ≤ 0.111 for all pairs).

**Figure 3 microorganisms-08-00106-f003:**
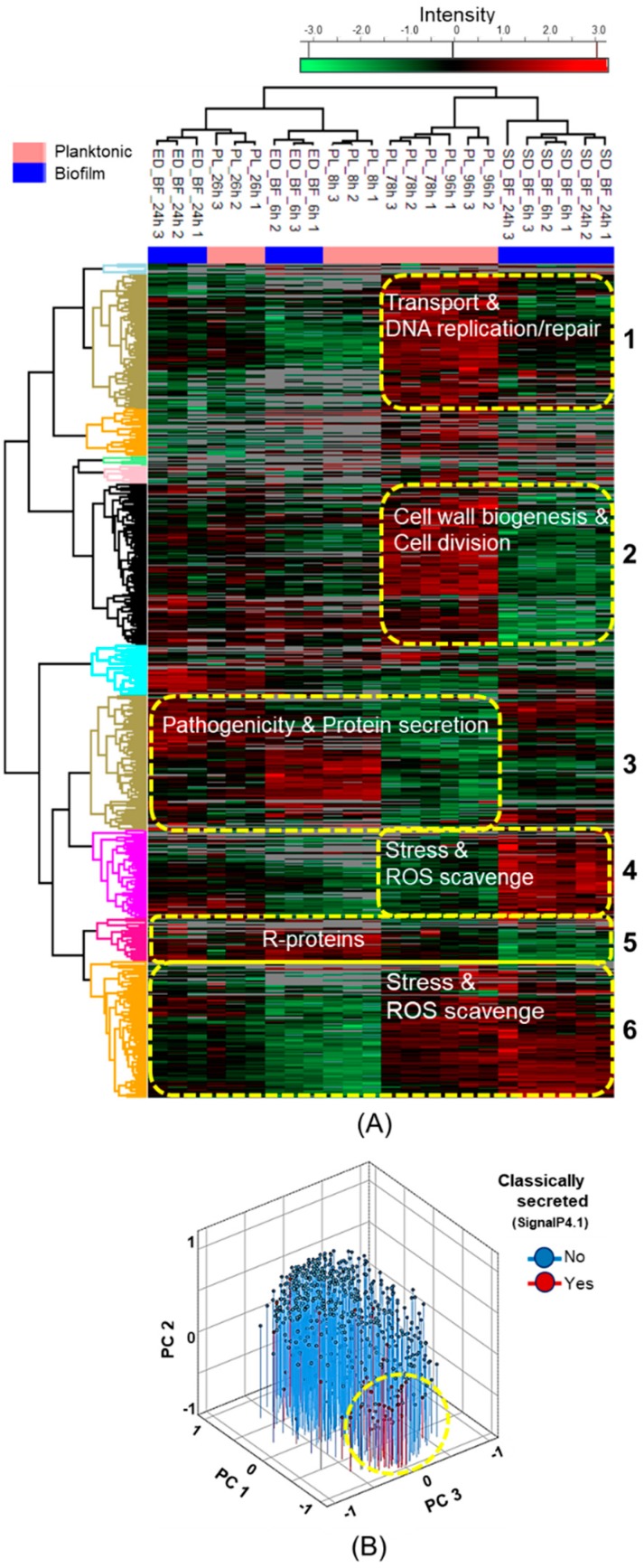
(**A**) Heatmap of row-wise standardized protein abundances (complete linkage, Lp norm; *p* = 1.5) with eleven clusters (distance threshold set at 50). Protein clusters with similar trends in specificity are circled. (**B**) 3D-PCA comparing protein abundance patterns with respect to their predicted secretion modes (predicted moonlighters are circled).

**Figure 4 microorganisms-08-00106-f004:**
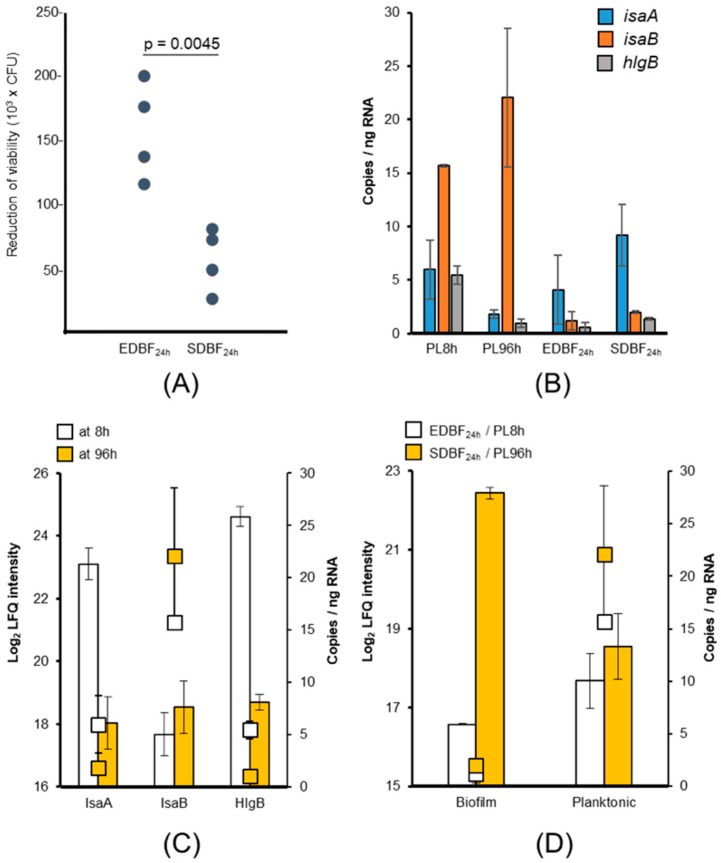
(**A**) THP-1-mediated phagocytosis/intra-cellular killing of the fully matured EDBF and SDBF cells at three and 24 hpi. Biofilm cells were detached from the polystyrene support and washed prior to the in vitro test. An unpaired *t*-test was used to compare the means of both groups. (**B**) Relative abundances of *isaA*, *isaB,* and *hlgB* specific transcripts in the PL8 h, PL96 h, EDBF_24 h, and SDBF_24 h cells. (**C**) Relative transcript (squares) versus the protein abundance (bars) levels for IsaA, IsaB, and HlgB in planktonic bacteria at 8 and 96 h time points. (**D**) Relative transcript (squares) versus protein abundance (bars) levels for IsaB during planktonic and biofilm growth. The error bars in panels B, C, and D indicate standard deviations for ≥2 protein label-free quantification (LFQ) intensity values and relative transcript abundances of two independent ddRT-PCR assays.

**Figure 5 microorganisms-08-00106-f005:**
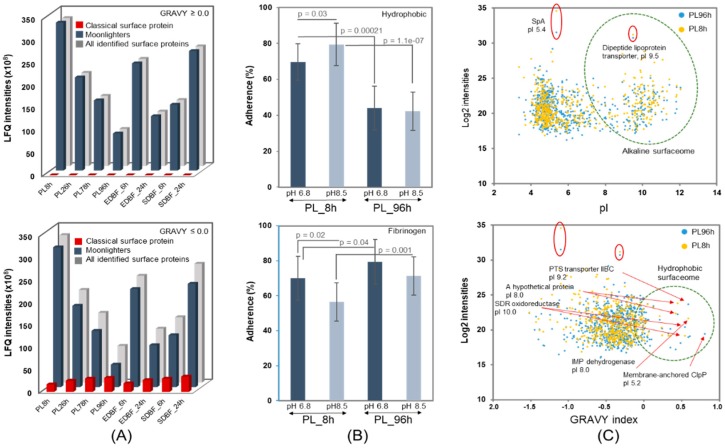
(**A**) Distribution of total LFQ intensities of all identified, moonlighters, and classically secreted proteins in indicated samples with Grand average of hydropathy (GRAVY) values ≥0.0 (upper panel) and ≤0.0 (lower panel). (**B**) Relative adherence of PL8 h and PL96 h cells to a hydrophobic material (upper panel). Bars in dark and light blue are cells washed with a pH 6.8 (*n* = 16) and pH 8.5 (*n* = 16) buffer, respectively. Relative adherence of PL8 h and PL96 h cells to human fibrinogen (lower panel). Bars in dark and light blue are cells washed with a pH 6.8 (*n* = 16) and pH 8.5 (*n* = 14) buffer. A paired *t*-test was used to compare the samples. (**C**) Proteins identified from PL8 h and PL96 h cells separated by their LFQ intensity (log2) and pI (upper panel) or GRAVY (lower panel) values. Alkaline and hydrophobic surfaceomes are circled. The most abundant proteins are circled, and highly hydrophobic proteins are marked with arrows.

**Table 1 microorganisms-08-00106-t001:** Abundance patterns of selected classical surface proteins on planktonic and biofilm cells.

Acc. No.	Virulence Factor Adhesin	Mr ^a^	pI ^a^	Gr ^a^	PL	EDBF	SDBF
8 h	26 h	78 h	96 h	6 h	24 h	6 h	24 h
WP_000594516.1	HlgAB subunit A	34.9	10.1	−0.6								
WP_001056917.1	HlgAB/HlgCB subunit B	36.8	9.7	−0.7								
WP_000595617.1	Gamma-hemolysin subunit B	38.7	9.0	−0.8								
WP_000669728.1	Protein MAP-domain containing	76.8	10.6	−0.5								
WP_000728713.1	Immunoglobulin G-binding—SpA	54.7	5.4	−1.1								
WP_000792567.1	Immunoglobulin-binding—Sbi	50.2	9.9	−1.0								
WP_000791398.1	Uncharacterized leukocidin	40.5	9.9	−1.1								
WP_000717395.1	Staphylococcal secretory SsaA2	29.6	9.1	−0.8								
WP_000745926.1	Clumping factor B ClfB	99.0	3.6	−1.2								
WP_077670278.1	Clumping factor A ClfA	101.3	3.4	−1.1								
WP_000751265.1	Immunodominant antigen IsaA	24.2	6.6	−0.3								
WP_001077096.1	Immunodominant antigen IsaB	19.5	10.0	−0.5								
WP_000769723.1	Peptidase propeptide YPEB	21.3	5.1	−1.1								
WP_000825534.1	Met ABC transporter	30.5	9.5	−0.3								
WP_001033875.1	5′-nucleotidase, lipoprotein e(P4) family	33.3	10.1	−0.8								
WP_001074521.1	Bifunctional autolysin—AtlA	137.9	10.1	−0.6								
WP_001170274.1	Autolysin—Sle1	36.0	10.0	−0.4								
WP_029051775.1	Extracellular matrix-binding—Ebh	1087.3	6.1	−0.7								
WP_038413132.1	Fibronectin-binding—FnbA	114.1	4.2	−0.9								
WP_038413163.1	Collagen adhesin	133.0	6.1	−1.0								
WP_001151905.1	Cell wall-anchored protein SasF	70.1	8.9	−0.7								
WP_000934494.1	MSCRAMM adhesin SdrD	145.2	4.0	−1.0								
WP_077670283.1	MSCRAMM adhesin SdrE (partial)	104.9	4.3	−0.9								
WP_077670284.1	MSCRAMM adhesin SdrC	88.5	4.6	−1.0								
WP_001041575.1	Heme uptake protein—IsdB	72.6	9.5	−0.9								
WP_000728052.1	Matrix binding adhesin Emp	38.5	10.5	−0.5								
WP_000782130.1	Foldase protein PrsA	35.6	9.8	−0.9								
WP_001229090.1	ABC transporter CntA	60.0	9.3	−0.8								
WP_000219068.1	Catabolite control—CcpA	36.0	6.4	−0.2								
WP_000737654.1	Manganese transport—MntC	34.7	9.1	−0.7								
										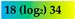

^a^ Mr, molecular mass/kDa; pI, isoelectric point; Gr, GRAVY index. PL, planktonic; EDBF and SDBF, biofilms initiated with exponential (2 h) and stationary phase cells (72 h), respectively. Color code, low (blue) and high (yellow) protein abundances. When cells are grey, identification is below detection limit.

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
