# Peer review of "Growth Mode and Physiological State of Cells Prior to Biofilm Formation Affect Immune Evasion and Persistence of Staphylococcus aureus"

_microorganisms, 2020, doi:10.3390/microorganisms8010106_

Round 1

Reviewer 1 Report

The manuscript addresses the analysis of cell surface proteins of S. aureus during planktonic and biofilm cultures. The authors performed a comparative analysis by LC-MS/MS to investigate all the surface proteins present in different growth conditions. Furthermore, the bacterial adhesion as well as phagocytosis by macrophages was investigated.  The authors analyzed the surfaceome in a growth-depend manner to reach high levofloxacin-tolerance. The manuscript is well written and easy to follow. However, I have only two questions/comments:

The statistical analysis is not explained. Please add the statistical tests used in this study in the method’s section. The name of the statistical test should be mentioned in the figures legends The complete study is done with one strain. How the authors argue that these results are extended to other staphylococcal strains? Is it possible to do few experiments at least with another non-related strain such as the intracellular killing by macrophage with EDF, SDF24h and planktonic cells and the real time pcr to check the abundances of isaA/B and hlgB? These results may improve the value of this manuscript.

Author Response

#Reviewer I

The manuscript addresses the analysis of cell surface proteins of S. aureus during planktonic and biofilm cultures. The authors performed a comparative analysis by LC-MS/MS to investigate all the surface proteins present in different growth conditions. Furthermore, the bacterial adhesion as well as phagocytosis by macrophages was investigated.  The authors analyzed the surfaceome in a growth-depend manner to reach high levofloxacin-tolerance. The manuscript is well written and easy to follow. However, I have only two questions/comments:

The statistical analysis is not explained. Please add the statistical tests used in this study in the method’s section. The name of the statistical test should be mentioned in the figures legends.

Our response: We agree and apologize for not being clear enough with statistics. We have now added the requested information in appropriate places in Materials and Methods and Figure legends.

The complete study is done with one strain. How the authors argue that these results are extended to other staphylococcal strains? Is it possible to do few experiments at least with another non-related strain such as the intracellular killing by macrophage with EDF, SDF24h and planktonic cells and the real time pcr to check the abundances of isaA/B and hlgB? These results may improve the value of this manuscript.

Our response: We agree that complementing these findings with other S. aureus strains would definitely add more value to our study and strengthen the hypothesis that prolonged cultivation and specific changes related to the indicated classical proteins could improve persistence and viability in vitro in staphylococci. The strain selected in our study was S. aureus subsp. aureus ATCC25923 that is a clinical isolate (designated as Seattle 1945) routinely used as a standard laboratory testing control strain in antibiotic susceptibility tests. This strain is also used as a biofilm model in numerous studies. We have considered the findings obtained with this model S. aureus as a starting point for subsequent studies, in which the impact of prolonged cultivation of different staphylococcus species/strains on in vitro viability will be examined using the same experimental setup. As S. aureus is a highly variable species, it could be that the increased viability does not necessarily apply to different species or even to closely related strains. In addition to the indicated classical surface proteins, known and novel moonlighters could also play a role in maintaining the viability of the cells after extended cultivation periods. More information related to this issue was added into the discussion.

Reviewer 2 Report

Authors explored that growth mode physiological state of cells prior to biofilm formation affect immune evasion and persistence of S. aureus. They exploited equivalent planktonic cells and biofilms of tationary-phase cells and exponential cells. The metabolic activities of biofilm and planktonic cells, immune evasion proteins, and levofloxacin-tolerance have been analyzed. It showed that the phenotypic states of S. aureus changed under different growth modes, associated with antibiotic tolerance. Generally, the manuscript is well designed and written.

However, it is unclear why only levofloxaxin was used. Can the conclusion be expanded for other antibiotics? Such phenomenon is universal or specific for selected antibiotic?  

There are too many keywords. Both the parts of Introduction and Discussion are too long and complicated, should be simplified.

L29: What is “THP-1 cell” tands for? It should be defined as a first time in abstract.

L97-99: “Biofilm seeding cultures were acquired……. with shaking…….”, was it acquired by static culture?

Statistical Analysis should be added in Materials and Methods.

The icons are not clear and the font is too large (A, B, C and D), the quality of the picture was need to improve in Figure 2 (the same for the others).

L478: Figure 6 does not appear in the manuscript, is it Figure 5?

Author Response

# Reviewer II

Authors explored that growth mode physiological state of cells prior to biofilm formation affect immune evasion and persistence of S. aureus. They exploited equivalent planktonic cells and biofilms of tationary-phase cells and exponential cells. The metabolic activities of biofilm and planktonic cells, immune evasion proteins, and levofloxacin-tolerance have been analyzed. It showed that the phenotypic states of S. aureus changed under different growth modes, associated with antibiotic tolerance. Generally, the manuscript is well designed and written.

However, it is unclear why only levofloxaxin was used. Can the conclusion be expanded for other antibiotics? Such phenomenon is universal or specific for selected antibiotic?

Our response: This is an important notion and we agree that persistence may present differently depending on the antimicrobial used. Many conventional antibiotics, such as beta-lactams, generally fail to kill non-growing bacteria regardless of other persistence-related traits, thus tolerance against them may not be strictly indicative of persistence as such. Fluoroquinolones, on the other hand, are more broadly active also against non-growing bacteria due to their ability to interfere with DNA replication. Levofloxacin is a broad-spectrum, third-generation fluoroquinolone antibiotic, used to treat bacterial infections, and was therefore chosen to validate that the antibiotic tolerance of our model aroused not only from halted growth and cell division, but also from a more complex and extensive phenotypic alteration. The matter is discussed further e.g. in Völzing KG & Brynildsen MP, mBio. 2015 Sep 1;6(5):e00731-15. doi: 10.1128/mBio.00731-15, and the selection of this antibiotic is added into the discussion.

There are too many keywords. Both the parts of Introduction and Discussion are too long and complicated, should be simplified.

Our response: We agree and have now corrected the number of keywords. We have also shortened and clarified the introduction and discussion (please, see the highlighted regions in both sections).

L29: What is “THP-1 cell” tands for? It should be defined as a first time in abstract.

Our response: THP-1 stands for the THP-1 macrophage/monocyte cells. This information is now added into the abstract.

L97-99: “Biofilm seeding cultures were acquired……. with shaking…….”, was it acquired by static culture?

Our response: We apologize for this mistake. The biofilms were created at 37C with shaking (220 rpm). This information is added into the method section.

Statistical Analysis should be added in Materials and Methods.

Our response: We agree and apologize for this mistake. Detailed information regarding the statistical analyses is added into relevant places in the methods section. The same information is also present in relevant Figure legends.  

The icons are not clear and the font is too large (A, B, C and D), the quality of the picture was need to improve in Figure 2 (the same for the others).

Our response: We agree and have modified and improved the quality of the figures.

L478: Figure 6 does not appear in the manuscript, is it Figure 5?

Our response: We apologize for this mistake; Figure 6 should be Figure 5. This is now corrected.

Reviewer 3 Report

Dear Authors,

The study is interesting and the manuscript is well written. I would like to make some suggestions.

Lines 478-490: Please check the number of figures.

Lines 714-716: It would be interesting to include a shorted version of this sentence in the conclusion in order to informe the readers about the potential influence of this experimental condition on the results.

Author Response

#Reviewer III

The study is interesting and the manuscript is well written. I would like to make some suggestions.

Lines 478-490: Please check the number of figures.

Our response: The number of the figures is checked.

Lines 714-716: It would be interesting to include a shorted version of this sentence in the conclusion in order to inform the readers about the potential influence of this experimental condition on the results.

Our response: We agree and have added the potential influence of CidC-mediated function on the results.